# Quasispecies Nature of RNA Viruses: Lessons from the Past

**DOI:** 10.3390/vaccines11020308

**Published:** 2023-01-30

**Authors:** Kiran Singh, Deepa Mehta, Shaurya Dumka, Aditya Singh Chauhan, Sachin Kumar

**Affiliations:** Department of Biosciences and Bioengineering, Indian Institute of Technology Guwahati, Guwahati 781039, India

**Keywords:** quasispecies, mutant cloud, lethal mutagenesis, antiviral treatments, vaccine

## Abstract

Viral quasispecies are distinct but closely related mutants formed by the disparity in viral genomes due to recombination, mutations, competition, and selection pressure. Theoretical derivation for the origin of a quasispecies is owed to the error-prone replication by polymerase and mutants of RNA replicators. Here, we briefly addressed the theoretical and mathematical origin of quasispecies and their dynamics. The impact of quasispecies for major salient human pathogens is reviewed. In the current global scenario, rapid changes in geographical landscapes favor the origin and selection of mutants. It comes as no surprise that a cauldron of mutants poses a significant risk to public health, capable of causing pandemics. Mutation rates in RNA viruses are magnitudes higher than in DNA organisms, explaining their enhanced virulence and evolvability. RNA viruses cause the most devastating pandemics; for example, members of the *Orthomyxoviridae* family caused the great influenza pandemic (1918 flu or Spanish flu), the SARS (severe acute respiratory syndrome) and MERS (Middle East respiratory syndrome) outbreak, and the human immunodeficiency viruses (HIV), lentiviruses of the *Retroviridae* family, caused worldwide devastation. Rapidly evolving RNA virus populations are a daunting challenge for the designing of effective control measures like vaccines. Developing awareness of the evolutionary dispositions of RNA viral mutant spectra and what influences their adaptation and virulence will help curtail outbreaks of past and future pathogens.

## 1. Preamble

Over the period of time, the term quasispecies in itself has evolved with many definitions, often interchangeably used to denote genetic variation within a population. If simplified, the quasispecies can be referred to as the diverse mutants in a population generated as a result of equilibria between mutation and natural selection [1,2,3,4]. RNA viruses constantly evolve, generating mutant swarms helping them to adapt to new environments. The generation of a quasispecies is an amalgamation of various mechanisms of genetic variation caused by mutation, gene duplication, recombination, and reassortment [5,6]. Low-fidelity RNA polymerases exhibit defective proofreading, limited template-copying ability, and ineffective post-replicative repair resulting in viral mutations [7]. The variations in the quasispecies pool are subjected to a continuous process of genetic variation, competition, and selection in the given environment for the survival of the fittest [7,8]. Quasispecies considerations are important for antiviral drug development and mechanistic studies.

### 1.1. Origin of Quasispecies

Eigen and Schuster, in 1970, explained the “precellular RNA world,” a mathematical framework to formulate the quasispecies theory [9,10]. Not only random mutations and fluctuations in the genome are responsible for quasispecies formation. Instead, variations may also occur by recombination and reassortment [11]. A few decades ago, quasispecies was considered as a simple genetic entity. Still, recent advancements in quasispecies theory have encouraged us to view and examine viruses as complex mutant spectra [8].

Though the concept of a quasispecies is taken as a single biological entity, its impact becomes more evident when the genome has high mutation rates in limited genome size (viz. RNA viruses) [7]. Whether viral or cellular, the quasispecies dynamics can operate in any of them due to higher mutation rates responsible for mutant spectra generation. When we talk of mutant spectra in RNA virus genetics, it does not involve only a single mutant but a cloud of mutants. Hence quasispecies is also termed mutant swarms or mutant clouds [7,12]. For the past three decades, the quasispecies theory has provided a population-based framework to understand RNA virus evolution [9]. Viral population load involved in the infections plays an important role in viral quasispecies evolution.

Francis Crick suggested and encouraged Manfred Eigen in a breakfast discussion to work on the first quantitative/mathematical treatment of replication systems, which undergoes a production of regular error copies [13,14]. These error copies were first termed ‘comet tail’ by Eigen, and later as quasispecies. Based on the macromolecular organization, replication, and adaptability, Eigen proposed a theory of the origin of life termed the quasispecies theory [13]. This theory was further taken ahead by Schuster and Eigen, and it was found to have an important application in the RNA virus’s evolution [13]. The concept of quasispecies arose by formulating these two fundamental mathematical equations: (a) concentration of mutant types as a function of replication time, and (b) the error threshold relationship [7,15]. These mathematical expressions of the quasispecies theory are explained in Figure 1. Charles Weissmann, in 1970, first gave an insight into the presence of mutant concentration as a function of replication time [11] (first equation in Figure 1). The higher the number of mutations that accumulate, the higher the probability of extinction; this error threshold tells us the maximum error limit to maintain the survival balance of the genetic information [14,15] (second equation in Figure 1).

Based on population dynamics, quasispecies can be understood by the given diagram: Quasispecies formation depends on the mutation rates or the error rates in replication. If the virus replication rate is accurately perfect, the viral offspring will occupy the same sequence space as shown in Figure 2. However, a mutation in the wild-type virion results in imperfect replication, and due to several biological constraints, the selection of a particular strain over other strains in the quasispecies cloud is observed (Figure 2).

In evolutionary terms, all the possible mutations in a gene or amino acid are represented in sequence space. Naturally occurring mutations are present in minimal functional space. So, the theoretically possible region is larger than the practically possible region. Mutations in the sequence space correlate with the expansion of the space depending on the high mutation rate and the selection of the variants (Figure 3) [14,16].

### 1.2. Contributors of Quasispecies Formation

The ability of viruses to generate and maintain genetic diversity to circumvent rapidly evolving environments and hosts is controlled by various drivers, including mutation, reassortment, and recombination. Random mutations by the RNA-dependent RNA polymerase (RdRps) viral polymerase are the major driving force for genetic diversity in RNA viruses. These low-fidelity RNA polymerases exhibit mutation rates of roughly 10^−4^ mutations per nucleotide copied, which is greater than nearly all DNA-dependent DNA polymerases [17,18,19,20]. Another major source of mutations in the RNA genome is contributed by host editing enzyme families like Apolipoprotein B mRNA Editing Catalytic Polypeptide (APOBEC) and adenosine deaminase, which is RNA-specific (ADAR). APOBECs are cytidine deaminases, which are DNA editing proteins that play a role in viral defense mechanisms in the host cells and are also involved in the process that often induces lethal hyper-mutation leading to the formation of a defective mutated progeny. In the case of ADAR1, the conversion of adenosines to inosines, occurring on self-complementary and immune-activating defective genomes, causes viral RNA hyper-mutations. ADAR1 hyper-mutation disrupts the complementary segments, thus eliminating the double-stranded substrate for immune activation as observed in the influenza virus [21], measles virus, and Rift Valley fever virus [22,23,24]. Viruses exist as a cloud of related sequences rather than a defined contig [11,25,26]. Recombination involves the formation of new sequence combinations by shuffling genetic material between parental genomes during genome replication [27]. Reassortment, formally called pseudo-recombination, occurs in viruses having segmented genomes or when two related viruses co-infect a cell and entire genome segments are swapped [28]. The fact that segmented viruses consistently evolve within and across families suggests that reassortment benefits viral fitness and evolution. Reassortment results in viruses that might be genetically diverse enough to break resistance [29,30], cause novel symptoms [31,32,33], alter host range [34,35], and evolve into new species.

Cumulatively, mutations at the genome level correspond to the changes at the phenotypic level, which are naturally selected in the population towards increased fitness. The two types of natural selection (positive and negative/purifying selection) play an important role in the evolution of viral quasispecies. When a newly derived mutation has an advantage over the other mutations, it is termed as ‘positive selection’, whereas ‘negative selection’ removes the deleterious mutations or variants. Hence negative selection is also sometimes called purifying selection. The positive selection increases the beneficial mutations, and the deleterious mutations are exempted from the population by the negative or purifying selection. In this way, natural selection influences the formation of a new mutation or a quasispecies [36,37]. During the course of viral evolution, host-to-host transmission, within-host transmission, and single-cell evolution play a critical role in shaping the quasispecies dynamics. When a mutant virus is shifted to a new environment, it has to either use a strategy of co-evolving or out-competing the fittest. Although, trying to out-compete excessive bottlenecks could lead to deleterious mutations. This phenomenon is known as Muller’s ratchet. So, in order to survive, a virus must balance the mutation rate between adaptability and fitness losses [38]. Firstly, new variants emerge at individual host levels further to jump through host-to-host transmission. Multicellular organisms impose heterogenous micro-environment for evolution and selection. Each tissue type provides the replicating virus with different selective pressures affecting its spread, pathogenesis, and evolution [36]. Host-to-host transmission is dependent upon bottleneck events, resulting in allelic frequency drift in the viral population [39]. Average fitness losses occur due to repeated bottleneck events. In contrast, rapid fitness gains are in accordance due to the large population passages [12].

Rather than genomes acting independently, variants that populate an environment can act cooperatively or in competition, thus dynamically affecting allelic frequencies. Those mutations or its combination that are advantageous in a particular environment can be positively selected (complementing the system), becoming dominant in a mutant spectrum. In contrast, negative selection is the process by which genomes are eliminated from the mutant spectrum or fall in frequency. The result is a fine-tuned ensemble of variants with specific frequencies optimized for that particular environment. Movement into diverse territories, such as an encounter with the immune system or a virus spread to a new tissue type, results in the re-equilibration of alleles to fit that particular environment. Biological constraints are negative interactions imposed on the selection factor to control the over-expansion of the sequence space [7]. A viral genome carrying a mutation occupies the sequence space and prevails to achieve fitness in the biological environment. A low mutation rate accompanies a stable population, whereas a high mutation rate could lead to new variants, and if the error threshold is crossed, it may lead to extinction (Figure 4). The occurrence of spontaneous genetic variations in RNA viruses could affect viral propagation and its virulence, ultimately implicating significant changes to the fatality curve.

## 2. Unique Features of the RNA Genome

The mutability of RNA viruses is around a million-fold higher than standard mutation rates operating in cellular DNA and is an established facet for RNA viruses. As a result, they are considered the first link between the quasispecies and virology [8,15]. A negative relationship between genome size and mutation rates suggests why DNA viruses mutate slower than RNA viruses [40]. The mutation rate in RNA viruses depends upon the extent of adaptability, which is supposed to correlate with its size [41]. By maintaining a relatively low fidelity rate, an RNA virus can rapidly traverse diverse environments with relative ease; in contrast, a high-fidelity replicating entity (such as a eukaryote) would be trapped in that sequence space and unable to adapt. This concept explains why the accumulation of mutations is advantageous to a virus, provided it is below the threshold leading to extinction.

## 3. Footprints of Quasispecies Evolution of Major RNA Viruses

Throughout history, outbreaks of emerging and re-emerging viral diseases have been a threat globally. Pandemics caused by viruses such as Influenza, MERS, Ebola, Polio, and others have been a catastrophe to mankind [42]. A timeline of model virus species that has caused devastating outbreaks in the past is given in Figure 5. The impact of any variant in causing the pandemic is very vividly observed during COVID-19. We all have witnessed the first, second, and third waves (in India majorly) of the COVID-19 pandemic causing mortality all over the globe.

Additionally, mutations in the receptor binding domain (RBD) and N-terminal domain led to an alpha, beta, gamma, delta, and omicron variant [43]. This has exposed the struggle in vaccine development and shed some light on improving our understanding of quasispecies formation in RNA viruses for developing new preventives. So, we aim to discuss quasispecies in some major RNA viruses’ families.

### 3.1. Paramyxoviridae

The family *Paramyxoviridae* consists of three genera: *Paramyxovirus*, which includes the parainfluenza viruses and mumps virus; *Pneumovirus*, which includes respiratory syncytial virus; and *Morbillivirus*, which includes the measles virus [44]. Virions in the family are enveloped and can be spherical or pleomorphic, capable of producing filaments. The diameter is around 150 nm. The nucleocapsid contains a monopartite, single-stranded, negative-sense RNA genome and an RNA-directed RNA polymerase. The genome is non-segmented, 15–19 kilobases in length, encoding 6–10 genes [45]. The nucleocapsid is surrounded by a lipid envelope that is studded with 8–12 nm spikes of two different transmembrane glycoproteins. The activities of these surface glycoproteins are the primary cause of diversity in the family. Quasispecies dynamics of members of paramyxoviruses have been profusely recognized, with processes of mutation, competition, and selection occurring as part of the natural life cycle of the virus, with evidence of past recombination events [46].

Due to the reasons mentioned above, one would expect the virus to be antigenically unstable. However, in contrast to instability in the RNA virus’s genome, members of the *Paramyxoviridae* family are antigenically stable. The reason behind this spectacle can be posited as follows: The genome is non-segmented and thus cannot undergo genetic reassortment. The second reason relates to the idea of antigenic drift. Another major hypothesis behind the stability of the virus is that the mutation leads to a decrease or total loss of function, which would, in turn, cause the new virus to be less efficient. These mutant progenies struggle to survive compared to the wild strains and would eventually die out. A fundamental concept in virus research is that genomes are packaged into particles to spread in the extracellular milieu. In most viruses, individual genomes are packaged. However, the pleomorphic particles of the members of the *Paramyxoviridae* family often incorporate more than one genome, as deduced initially from particle sedimentation and ultraviolet inactivation studies [47], causing the formation of genetically divergent progeny.

Pneumoviruses are pleomorphic, enveloped virions ranging from 150 to 200 nm in diameter and capable of producing filaments [48]. Fatal effects of human respiratory syncytial virus (HRSV)-associated disease are seen in children below the age of 1 year, while bovine respiratory syncytial virus (BRSV)-associated illness is usually terminal in calves below the age of 6 months. HRSVs are classified into two major groups (A and B) based on antigenic differences in the G glycoprotein. [49]. BRSV and HRSV are closely related, but antigenic differences have been observed. The G glycoprotein in these viruses shows extreme antigenic and sequence divergence, with only 47% amino acid homology between prototype HRSV A and B viruses [49]. It has been hypothesized that changes in amino acid sequences benefit the virus allowing it to escape immunity by modifying its epitopes [50].

### 3.2. Orthomyxoviridae

Viruses belonging to the *Orthomyxoviridae* family are known as orthomyxoviruses. These are enveloped virions that measure between 80–120 nm in diameter, with nucleocapsids forming a helical symmetry [51]. These viruses possess segmented and negative-sense single-stranded RNA genomes. The genome consists of 7–8 RNA segments. These segments encode different structural and non-structural proteins. Some structural proteins include hemagglutinin (HA), neuraminidase (NA), two matrix proteins, and a nucleoprotein. Orthomyxoviruses cause respiratory problems in humans as well as in animals. *Orthomyxoviridae* family is divided into seven genera: Influenza viruses A, B, C, and D, Quaranjavirus, Thogotovirus, and Isavirus [52]. Infection in humans is generally caused by the Influenza A virus and is the one with pandemic potential [53]. Worldwide, influenza causes 500,000 deaths, and 3–5 million people are affected severely [54]. The significant proteins that undergo mutations are HA and NA structural surface glycoproteins. HA and NA proteins are substantial targets for neutralizing antibodies the host immune system produces. Since these proteins are considered to show high mutability, which becomes a significant limitation for antiviral therapy [55], due to the segmented viral genome, during maturation virus may undergo reassortments resulting in different combinations of genes causing the emergence of a new virus subtype [56]. As per the reports in 2019, nearly 1–18 HAs and 1–11 NAs are involved in viral attachment and release [57]. If reassortment occurs between these 18 HAs and 11 NAs, a total of 198 combinations are possible. In humans, four combinations of 3 HAs and 3 NAs have been found, which are H1N1, H2N2, H3N2, and possibly H3N8. However, mutability and reassortment are two different and distinct phenomena but are the major contributing factors to the overall variability of the influenza virus.

#### 3.2.1. Avian Influenza

Based on the virulence potential in chickens, avian influenza is divided into two categories: low pathogenic avian influenza (LPAI) and highly pathogenic avian influenza (HPAI). Due to constant changes in the gene arrangements of influenza viruses in different species, especially in human-animal interaction, many avian influenza viruses are responsible for causing human outbreaks [58]. H5N1 was the first HPAI virus that caused a pandemic in Hongkong in the year 1997 [59]. According to researchers and many historians, the first influenza pandemic likely occurred in 1510 [60]. The first well-known pandemic was the Russian flu, which occurred in 1889 and 1893, caused by IAV/H3N8 based on the data [61,62]. Spanish flu caused by IAV/H1N1 came into existence in 1918 and arose from an avian influenza virus [63], resulting in high morbidity and low mortality. This time the death rate rose by 50 folds compared to the Russian flu pandemic, and over 50 million people lost their lives [64]. Gene reassortments among humans, avian, and swine IAV/H1N1 of the 1918 pandemic are considered the root cause of the 1957, 1968, and 2009 pandemics [65]. Asian flu of 1957–1959, caused by the IAV/H2N2, was another pandemic the world saw [66]. Approximately 1–2 million deaths occurred during this period. It was only less than a decade when the Hongkong flu came into existence in 1968–1970, caused by a new combination of IAV [66]. The 2009 pandemic occurred due to the triple reassortment of human, avian, and swine influenza virus and gave rise to IAV/H1N1 [67]. Overall, the influenza pandemic depends on virulence and the power to reassort and form new combinations by genetic drift and genetic shift, which poses a significant challenge to tackling the pandemics each time [68].

#### 3.2.2. Mutations in Influenza Leading toward Quasispecies

The two possible reasons for the influenza viruses’ evolution are antigenic drift and antigenic shift. Mutations in the HA and NA gene of influenza is responsible for antigenic drift. These mutations are ultimately responsible for different HA and NA protein combinations and cause pandemics [58]. This antigenic drift is present in all influenza strains and is highest in A, followed by B, C, and D [69]. The antigenic shift does not occur with the different genera of influenza viruses and is very common among Influenza A Viruses (IAVs) [69,70].

Different combinations of strains are present in the environment, and a pool of viral variants generated from a common species because of mutation or reassortment when undergoing competition and selection gives rise to quasispecies, as mentioned above. This is known to cause pandemics from time to time. So, it becomes challenging to predict what and when will the next combination come into existence [14,71]. On the one hand, this is a big obstacle for the virologists, while on the other, it also becomes a potential opportunity to work on this.

### 3.3. Reteroviridae

The retrovirus family is widespread and found in varieties of vertebrate hosts. These are enveloped virions that measure between 80–100 nm. The members of this family contain linear, positive sense single-stranded RNA and reverse transcriptase enzyme, which reverse transcribes its genome into DNA. The reverse transcribed DNA is integrated into the host DNA and gives rise to the provirus. If the viral DNA integration occurs into the germline tissue, then it can give rise to a heritable provirus known as an endogenous retrovirus [72].

Retrovirus family virion contains four genes 5′-gag-pro-pol-env-3′, coding for various viral proteins. The env gene of retrovirus encodes for two types of proteins that is a surface protein (SU) and transmembrane protein (TU). The gag gene of retrovirus encodes the matrix protein (MA), capsid protein (CA), and nucleocapsid protein (NA). The pol gene of the retroviral family virion is responsible for synthesizing reverse transcriptase (RT) and integrase [73]. The lack of proofreading activity and mechanism of viral reverse transcriptase enzyme is responsible for the high rate of genetic diversity and recombination and accountable for the formation of quasispecies or mutant swarm populations of the retroviruses [74].

The retrovirus family virion is responsible for many diseases, such as immunodeficiency (AIDS), autoimmune disease, lower motor neuron diseases, and several acute diseases, including tissue damage. Some of the retroviruses are also associated with malignancies, including lymphoma, sarcoma, and certain leukemia. Retroviruses can be transmitted via both horizontal and direct routes. The horizontal transmission includes transmission via blood, saliva, sexual contact, etc., whereas the direct routes involve the infection of the developing embryos or via milk, etc. [75]. The HIV-1 virus is a member of the retrovirus family with exhibits characteristics that increase its potency to form quasispecies. The heterogeneity and diversification of viral sequences in the HIV-1 population are due to the high mutation rate, short generation time, virion production rate, and high frequency of recombination that contributes to quasispecies dynamics [72,73,74]. The heterogeneity and diversification of viral sequence are more complicated by the reactivation of the latent provirus, compartmentalization of the infection, and the presence of a viral reservoir [75,76,77,78]. The proviral activation will contribute to new replicating sequences, thus generating millions of recombinant and mutant viruses [79,80].

HIV-1 virus strains are classified into two types that are R5- tropic virus strain and X4-tropic virus strain, depending on the type of coreceptor they used to infect the target cell. Generally, R5- tropic virus strains are found in the early stage of the disease, whereas X4-tropic virus strains are predominant during the late phase of the disease [81,82]. A classic example of HIV X4 selection is evidenced in a case report on a patient from Berlin, who was suffering from HIV and acute myeloid leukemia. The patient was given HAART therapy with allogenic stem-cell transplantation, homozygous for the CCR5 variant. The transplantation was successful for R5 elimination, but it led to the selection of the CXCR4- tropic variant (X4), which used the coreceptor [83]. R5-strain can induce CD4-T cell apoptosis in the late phase of the disease. X4-strain of virus can infect naive T cells, which is likely to be responsible for the association of the X4-strain virus with the disease progression [84,85].

The capacity of replication or “fitness” is the key parameter of virion that influences HIV-1 behavior and response to selective constraints [86]. Multiple mechanisms of HIV-1 RNA interference and drug resistance have been described [86,87]. The input of new genomic sequences, high mutation rate, and mutation that diminishes the sensitivity to the antiretroviral agent is responsible for drug-resistant [88,89]. Recombination can also be responsible for generating multi-drug resistant viral strains by bringing together different genomic sequences that each provide resistance to antiretroviral drugs [90,91]. Only partially suppressed replication and suboptimal treatment are also responsible for the rapid development of drug-resistant [92,93]. The host immune system imposes multiple selective constrain that act upon all viral infections. In the case of HIV-1, there is an added complication in that the immune cell itself gets infected, causing immunosenescence and thereby modifying the immunological environment [94]. HIV-1 has developed both interaction and evasion strategies to cope with the immune response. The escape mutant of HIV-1 contributes to virus survival in response to neutralizing antibodies and cytotoxic T-cells [95]. The functional impairment of HIV-1 specific CD8-T cells and continuous viral replication is responsible for uncontrolled HIV-1 infection, and that is also the reason why HIV-1 often emerges as the winner in the arms race between the viral quasispecies and immune system [96,97]. The ability of HIV-1 to evade the host immune response and persistence for long periods constitute a major difficulty for the design of an effective vaccine.

### 3.4. Coronaviridae

Coronaviruses with their signature crown-shaped capsid structure belong to the family *Coronaviridae*; they are classified based on their host choice; alpha and beta coronaviruses, infecting mammals, and gamma coronaviruses, infecting birds. These are the largest, positive-sense single-stranded RNA viruses, whose 31 kb genome comprises structural glycoprotein including spike protein (S), an envelope protein (E), glycoprotein (M), and nucleocapsid (N) and accessory regions (ORFs) [42].

Coronaviruses were less of a concern until 2002, when the SARS outbreak occurred [98]. The transmission of the SARS-CoV-related virus was first observed in horseshoe bats as their primordial hosts and palm civets as intermediate hosts. Dogs, cats, and mice act as a reservoir for human coronaviruses (HCoVs). Before 2002 these were associated with mild to severe respiratory concerns in immune-compromised individuals. After SARS, MERS CoV caused an outbreak in Saudi Arabia, affecting 27 countries [99] and giving implications to mutation trajectories encasing the whole population. MERS CoV has zoonotic implications from camels to humans, transmitted from direct and indirect contact with camels and contaminated products [100]. In 2019, SARS-CoV caused a pandemic with severe infections causing high mortality [101,102]. The transmissibility of SARS-CoV-2 is highest in comparison to other SARS-CoV and influenza viruses [103].

MERS-CoV has the largest size among all coronaviruses, i.e., ~30.11 kb, compared to SARS-CoV-2 and SARS-CoV, both of ~29 Kb. All coronaviruses mediate entry by attaching the receptor-binding domain to different host surface receptors. For SARS-CoV, angiotensin-converting enzyme 2 (ACE 2) acts as the primary receptor, but it also facilities it’s entry via CD209 and CD209L. MERS-CoV uses CD26 as an entry receptor. SARS-CoV-2 shares ACE 2 with SARS-CoV along with integrins to make an entry in the cell [104]. However, SARS-CoV-2 has a higher affinity for ACE-2 than SAR-CoV. This diversity in targeting the host cell receptor suggests selective mutations to increase the host range’s adaptivity and expansion. The genomic variation that arose due to inter- and intra-host evolution has led to MERS-CoV, SARS-CoV, and SARS-CoV-2 with diverse host ranges [100].

Some experimental evidence has shown that the deviation from the wild-type mutation has significantly affected the viral kinetics and its infectivity. As in the case of double mutations in (A9G/R13A) in the non-structural protein 1a (nsp1) of the MERS coronavirus causes lower infectivity and smaller plaque size compared to wild-type viruses [105]. Also, variations in the coronavirus spike glycoprotein due to natural and experimentally induced mutations give rise to changed viral tropism and pathogenesis.

A variant of wild-type MERS-CoV, comprising an out-of-frame deletion of 530 nucleotides in the spike glycoprotein gene, caused the loss of a large portion of the S2 subunit resulting in the production of defective particles with low infectivity. However, the mutation did provide aid to the wild-type MERS-CoV infection by producing a misfolded S1 protein, which acted as a trap toward the spike-specific neutralizing antibodies [106].

Another study analyzing non-consensus sequences from 24 patients infected with MERS-CoV revealed high-level heterogeneity among samples [107]. After analyzing the samples, mutations were found in the receptor-binding domain (RBD) of the viral spike glycoprotein. These mutations resulted in the low affinity of RBD for dipeptidyl peptidase-4 (CD26), rendering viral fitness. Although, the frequency of wild type also gets reduced to approximately 10% in all the samples, suggesting selection enforced by the host immune system on the genetic variants. The effect of spike glycoprotein mutation in T1015N of MERS-CoV on its propagation and plaque morphology in vitro is supported by an increased replication rate of 0.5 logs and the formation of bigger plaques than wild-type MERS-CoV [108].

Analysis of quasispecies in SARS-CoV-1 from 9 patients showed nine variable sites among 107 variations [109]. The quasispecies in two samples from SARS-CoV-2 patients from Italy revealed the two nucleotide mutations A-T and G-A substitutions in the ORF 1ab gene at 2269 and 7388, respectively [110]. Clearly, analysis of variants from clinical samples arose de novo and could suggest host-pathogen relationships and adaptability in various environments. An understanding of which mutation could cost viral fitness can be a benefit.

### 3.5. Other Viruses

Concerns related to global threats due to emerging and remerging viruses are increasing day by day. Apart from the families discussed above, some other families like *Picornaviridae* (Polio), *Togaviridae* (Chikungunya), *Flaviviridae* (Hepatitis C virus), and *Filoviridae* (Ebola) have imposed significant mortality in public. The evidence of quasispecies formation in some of the important RNA viruses is reviewed in the following section.

#### 3.5.1. Hepatitis Virus

This class includes a range of unrelated human pathogens like the Hepatitis A virus (HAV), an unenveloped RNA virus belonging to the *Picornaviridae* family; the Hepatitis C virus (HCV), a single-stranded enveloped RNA virus belonging to *Flaviviridae*; the Hepatitis D virus (HDV), single-stranded circular RNA virus belonging to Hepadna viruses, and Hepatitis E virus (HEV), another non-enveloped single-stranded RNA virus belonging to Caliciviruses [111]. Among these, HCV acts as a prototype of quasispecies formation. It was first identified in 1989, and since then, several genotypes of HCV have been discovered. HCV is associated with blood transfusions and spreads to the liver resulting in cirrhosis and hepatic carcinoma. Approximately 58 million people annually develop chronic HCV infection. HCV shares feature with other members of the *Flaviviridae* family, like Dengue fever virus, Japanese Encephalitis virus, and Classical swine fever virus. All these members have enveloped virions containing positive sense, single-stranded RNA genomes of 9.6 to 12.3 kb. Encoding for structural and non-structural proteins. HCV has a gene NS5B for the RNA-dependent RNA polymerase that lacks proofreading activity. It incorporates nucleotides with a high error rate of 10^−3^ errors/site, which leads to heterogeneity in the gnome. Another factor responsible for quasispecies formation is the hypervariable region (HVR) in the envelope protein (E2). HVR-1 is located at the N-terminal of E2, while HVR-2 is slightly downstream of HVR-1. HVR-1 acts as a dominant epitope for neutralizing antibodies. Mutations in the HVR-1 are responsible for the formation of escape mutants under host immune selection pressure. HVR-1 and HVR-2 variations are responsible for virus tropism, virulence, and resistance against drugs. During the acute phase of infection, HCV has a higher rate of mismatched substitution per site. One case study suggests that the HVR-1 does not display neutralizing epitopes during infection. The existence of various closely related mutants during the acute phase and their adaptation leads to chronic infection [112]. Other mutations contributing to quasispecies spectra are phosphorylated non-structural proteinNS5A, which modulates RNA-dependent RNA polymerase activity. It is also attributed to the attenuation of interferon (IFN) activity. Mutations are reported in the IFN-sensitivity-determining region (ISDR) of NS5A that often affects IFN therapy [113].

#### 3.5.2. Ebolavirus

The Ebola virus belongs to the *Filoviridae* family, which includes two other genera: Cuevavirus and Marburgvirus [114]. When viewed under an electron microscope, the virion shape resembles a twisted thread; hence the family has been named *filoviridae* because, in Latin, ‘filum’ means ‘thread’ [114]. It is a linear, non-segmented, single-stranded, negative-sense RNA. There is a long historical background of hemorrhagic fever. First recorded in 1944, the Crimean-Congo hemorrhagic fever was later spread to Korea, Argentinian, Bolivian, and Lassa between 1955–1969. It was 1976 when the first cases of Ebola-borne hemorrhagic fever (now called Ebola virus disease) were recorded in the Democratic Republic of Congo (formerly Zaire). According to a WHO report, the fatality rate of Ebola virus disease (EVD) is approximately 50%. Because of its RNA nature, Ebola Virus has the potential to emerge as the next pandemic in the world, and its different strains lead to a quasispecies formation. In the case of Ebola, virus tropism is due to glycoprotein (GP) mutation. It was evidenced in the case study, where sequences of the glycoprotein of 66 Ebola isolates were aligned from the old (1976 to 2005), and new (2014) outbreaks showed variations in the region. Nucleotide mutations at two positions, A82V and P382T were observed in the new isolates [115]. This accounts for the emergence of the 2014 variant from the pool of quasispecies having mutations in the GP region. In some other studies, mutations at positions A82V and T544I has produced virus with increased infectivity [116,117,118].

#### 3.5.3. Poliovirus

Poliovirus, which belongs to the *Picornaviridae* family, causes poliomyelitis. It is a non-enveloped, positive-sense single-stranded RNA (+ssRNA) virus [42]. The virion has an icosahedral symmetry and is 27 nm in diameter comprising 7500 nucleotides. Being an RNA virus, it has relatively high mutation rates. There are three poliovirus serotypes: 1, 2, and 3, out of which 1 is the most common form in nature. The two mechanisms that help the virus to evade the immune system are, first, to replicate very quickly before mounting an immune response and second, to survive the highly acidic conditions of the stomach. Polio has been significantly reduced all over the world by WHO 1988 eradication program, but certain countries are still affected (WHO, 2019). A study was conducted on poliovirus to understand the effect of limiting genomic diversity on the evolution of the virus. A substitution of Glycine at position 64 (G64S) in the polymerase was done. Here the mutant showed higher fidelity and a noticeable effect on adaptation and pathogenicity [119]. Another study revealed that poliovirus infection in mice has diversity in tissue-specific patterns within individual organs [120]. Sequencing results from the spleen, kidney, and liver showed distinct quasispecies populations across individual mice.

#### 3.5.4. Chikungunya Virus

Chikungunya (CHIK), which belongs to the family *Togaviridae*, is an arthropod-borne virus (arboviruses) that causes high morbidity in humans; it infects humans via mosquitos [121]. It has some recurrent past events, majorly affecting Africa, Southeast Asia, and the 2013 outbreak in America [122]. Within the host and population-based selection, pressure for quasispecies development is highly pronounced in the arboviruses. A study in the murine model investigated low and high-fidelity mutants’ genetic diversity with the host [123]. Both high- and low-fidelity mutants showed low virulence compared to the wild type. The high-fidelity mutants reported varied diversification in NGS data. Increased virulence is observed due to mutation in nsp2 G641D and nsp4 C483Y of high-fidelity mutants. CHIK variants showed different plaque morphology, reported in 2005 [124]. Small plaque sizes show reduced in-vivo fitness variants and are maintained in natural quasispecies in the viral pool.

Till now, we have discussed and analyzed how specific mutations occur in the viral genome and how their selection is influenced by environmental and host-pathogen interactions, which could affect the replication rate in the host. Understanding replication rate, plaque morphology, and sequence analysis could shed light on the intricate phenomenon of quasispecies selection and might provide evidence for resolving obstacles to developing therapies in controlling current and future pandemics.

## 4. Challenges and Discussions for Prospective Antiviral Treatments

Viral quasispecies under selection pressure led to the formation of escape mutants; these low-frequency strains escape immune targeting and proliferate in the host system. This is the biggest challenge in developing vaccines against RNA viruses [94,125,126]. Escape from antiviral drugs leads to vaccine failures and fatality. The selection of drug-resistant mutants from the inhibitory effects of antiviral drugs is another big challenge. To unveil new antiviral preventive measures, we must comprehend the interplay of the escape mutants and their co-evolution strategies.

In the case of influenza, mutants have shown resistance against administered drugs Amantadine and Rimantadine [127]. Many mutations are observed in influenza strains due to the use of the anti-influenza drug oseltamivir [128]. These resistant strains of influenza are found in bird populations, which could give rise to emerging strains of influenza. Nevertheless, the collective fitness loss of resistant species to specific drugs in combinational therapy confirms the replication of the virus up to minimized levels. Hence, proving the effectiveness of combinational therapy against low-frequency undetected mutants.

RNA interference (RNAi) systems are a promising alternative to antiviral drugs. Synthetic miRNA (siRNA) targeting specific viral sequences [129] leads to BHK cell survival when infected with Dengue virus 2 (DENV2) and significantly reduces virus titer [130]. Also, inhibition was improved when delivery of siRNA was replaced with lentivirus. Some examples hold evidence for the effectiveness of RNAi, like, RNAi against ST6GAL1 resulted in the inhibition of binding of influenza to the sialic receptor, hence preventing infection [131]. In HIV, siRNA against CCR5 led to the inhibition of HIV infection in RNAi-treated macrophages [130]. Although, careful consideration is needed while designing the RNAi as it may cause toxic, harmful effects, rendering its efficiency. Another concern is that resistant mutants may rise due to treatment with RNAi, which could be the dominant strain under selection pressure. RNAi escape mutants are seen to have mutations in the regions targeted by siRNA evidence found in poliovirus, JEV, Hepatitis C viruses, and others [132,133,134].

Another important treatment is lethal mutagenesis, which also holds some promising results. Administration of mutagenic pyrimidine analog in AIDS patients resulted in lethal mutagenesis [135]. When a non-mutagenic and mutagenic agent is administered in therapy, it results in escape mutants which either complement or interfere with the system; however, interference with the replication and infectivity of the wild-type genome results in the replication system breakdown. Some commonly used mutagenic agents are tabulated in Table 1. Additionally, virus extinction is strongly controlled by mutagenesis and defective interfering genomes [13]. As the ratio of mutants increases, it evokes an extreme synergetic interference in the growth kinetics of the virus. One such piece of evidence is the delayed replication of drug-resistant poliovirus mutants in the presence of trans-acting mutants [136,137]. In the presence of mutagens like ribavirin (a nucleoside analog), FMDV mutants experience interference due to the generation of some defector genomes obstructing replication. Also, inhibitors like guanidine hydrochloride (GU) produce a suppressive effect on the interference caused due to defector mutants on wild-type proliferation [138].

Even a mutation in a single amino acid could give rise to enough interference to convert a genome into an interfering genome (the reverse is true as well), although the probability of mutation in the sequence space is random. Accordingly, this can be induced in-vitro by increasing the viral load of defective interfering particles in the system [139,140]. This could provide insight into developing new therapeutics against mutants that interfere with wild-type variants. The spectrum of variants arose due to the complementation and interference of altered viral proteins affecting the antiviral treatments. Understanding the role of these interfering particles (defective interfering particles) could lead to the successful development of antiviral therapy.

Consequently, we can address the challenge that arose due to the quasispecies selection of the dominant variant by sequential or combination treatment involving alternative inhibitors and mutagenic agents in clinical trials [135]. There are some theoretical and experimental shreds of evidences of the benefits of sequential therapy over combinational, which could result in the extinction of a mutagen-resistant variant [141,142]. There is a need to intensively comprehend combinational therapies as it holds promise to develop minimized virus load, and the critically analyzed siRNA or lethal mutagens could possibly prevent the fitness cost of escape mutants.

## 5. Concluding Remarks

In experimental virology, the physiochemical milieu affects the frequency of mutants and their nature, while in-vivo selection and biological constraint interfere with the quasispecies formation [15]. Selection and survival solely depend on the extent of the accumulation of the favorable mutation. Transmission between individual host and host-to-host evolution shows that heterogeneity in the genome arises from various factors. Interestingly, the extinction of virus species as a cost of a high mutation rate has importance to virologists. It could help if we could design the drug to manipulate the virus mutation rates, as high mutation rates can push the population beyond the error threshold. Even though the hyper-mutability of RNA viruses works in our favor, viruses manage to conserve their identity in terms of genetic robustness to maintain the viral gene pool [16]. As we have reviewed above, many factors contribute to the efficacy of antiviral drugs, including quasispecies dynamics. Evolution differs for heterogeneous tissue/organs, even within the same individual virus. Tissue-specific micro-environment provides a different set of parameters for virus replication. So, the development of effective drugs and vaccines should encircle all factors in order to be effective against mutants, as the generation of novel mutants with higher pathogenicity and transmissibility is vividly seen in the case of COVID-19. These escape mutants are selected under the pressure of antiviral drugs and host immune response. To control the generation of escape mutants, we must understand the effect of combination therapies for each step of the virus life cycle, which could help eliminate the probability of the generation of variants. Understanding the upper limit constraint of virus replication and the effectiveness of combinational therapies, which leads to the extinction of the virus, is the key to solving the mystery of antiviral treatments [7].

## Figures and Tables

**Figure 1 vaccines-11-00308-f001:**
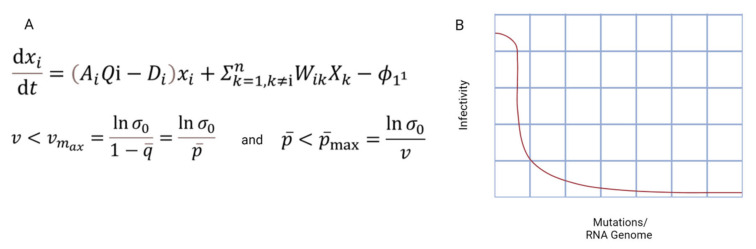
Quasispecies theory is described by using two fundamental equations. (**A**) The first equation describes the change in the concentration of mutant i with respect to time, xi(t); similarly, xk(t) describes the concentration of mutant k. The second equation expresses the error threshold relationship. The second equation has importance in virology and lethal mutagenesis, where the mutation rate pushes the virus beyond the error threshold and has benefits in drug development. (**B**) Error Threshold: the higher the mutations, the lower the infectivity.

**Figure 2 vaccines-11-00308-f002:**
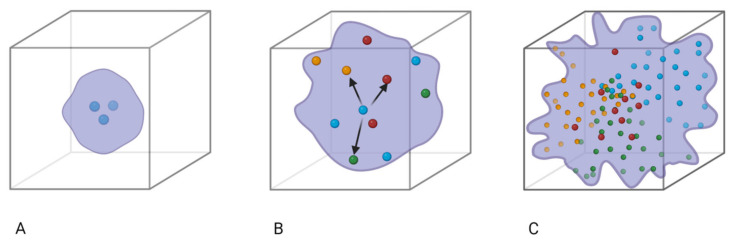
(**A**) Model depicting wild-type virion with allelic differences. (**B**) Model showing a mutation in the wild-type virion (yellow, red, and green). (**C**) Model showing the quasispecies cloud and selection of a particular strain over other strains due to biological constraints.

**Figure 3 vaccines-11-00308-f003:**
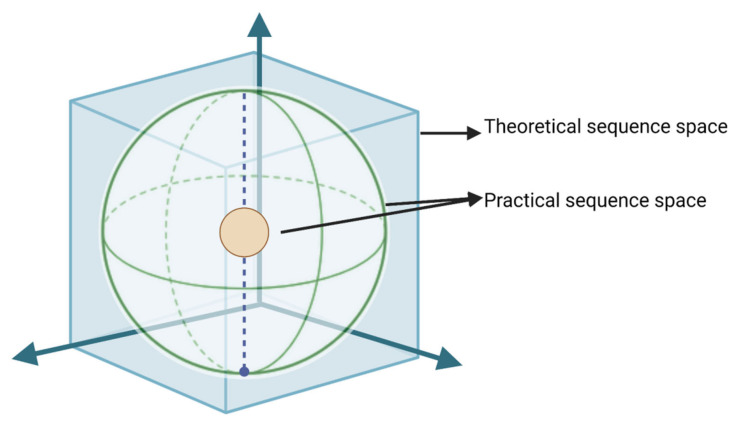
This is a simplified model of a 3D representation of the theoretical sequence space possible and available sequence space for viruses. Here, the square depicts the theoretical sequence space. The sphere represents the occupancy of virus mutants. This 10-fold difference in the smaller and bigger sphere explains the difference in the available occupancy of two viruses.

**Figure 4 vaccines-11-00308-f004:**
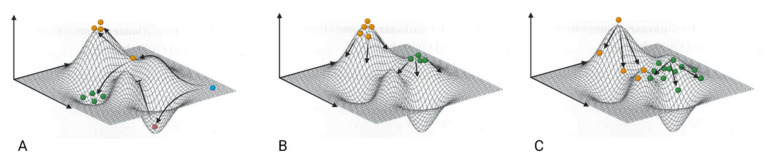
(**A**) The fitness landscape represents a relationship between genotype and evolutionary fitness. It allows the mutation to settle and thrive in the sequence space. Here, peaks and valleys represent high fitness due to the accumulation of specific mutations and low fitness due to minor fitness losses, respectively. A typical wild-type virus (represented as blue) that tries to occupy sequence space of the nearest peak by consecutive mutations will gain fitness and survive. At the same time, if it tries to occupy the sequence space of the valley, it will lead to the extinction of the mutant type (red). While in certain scenarios, mutations could lead to comparatively decreased fitness (green), so the virus is not extinct but incapable of proliferation. (**B**) In the above figure, mutants with low mutation rates are more stable (green) and cluster at the neighboring peaks, but the fittest will out compete the others. Mutants with high mutation rates will try to occupy neighboring peaks but, due to large mutational shifts, get trapped in the sequence space and will not spread out (yellow). (**C**) In order to achieve the highest peak, wild jumps are needed, which consequently leads to fitness and, therefore, the probability is low for their survival. On the other hand, mutants with slow and stable leaps are mutationally robust (green ones on comparatively flatter peaks) and will prevail. Therefore, the mutants on the higher peak (yellow) will always represent lower mean fitness in comparison to mutants on flatter peaks (green).

**Figure 5 vaccines-11-00308-f005:**
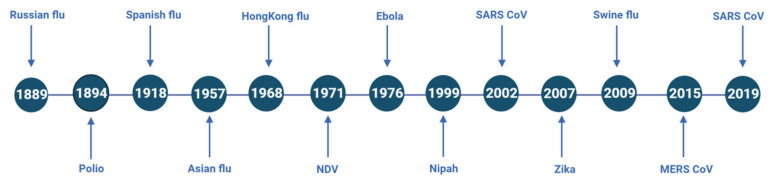
Chronological representation of pandemics that occurred in the past.

**Table 1 vaccines-11-00308-t001:** A list of mutagenic base and nucleotide analogs used in lethal mutagenesis of RNA viruses.

Virus	Family	Mutagenic Base and Nucleotide Analog
Polio virus	*Picornaviridae*	Ribavirin, 5-nitrocytidine
Dengue virus	*Flavivirus*	5-Fluorouracil, ribavirin
Zika virus	*Flavivirus*	Ribavirin, favipiravir
Hepatitis C virus	*Flaviviridae*	Ribavirin, favipiravir
Hepatitis E virus	*Hepeviridae*	Ribavirin
SARS-CoV-2	*Coronaviridae*	Favipiravir
Tobacco mosaic virus	*Virgaviridae*	5-fluorouracil
Influenza A virus	*Orthomyxoviridae*	5-fluorouracil, Ribavirin 5-azacytidine
Vesicular stomatitis virus	*Rhabdoviridae*	5-fluorouracil
Hantaan virus	*Hantaviridae*	Ribavirin
Rift valley fever virus	*Phenuiviridae*	Favipiravir
Ebola virus	*Filoviridae*	5-fluorouracil
Marburg virus	*Filovirade*	Favipiravir

## Data Availability

Not applicable.

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
