# Peer review of "Quasispecies Nature of RNA Viruses: Lessons from the Past"

_vaccines, 2023, doi:10.3390/vaccines11020308_

Round 1

Reviewer 1 Report

In the manuscript, the authors provide a review of quasispecies development during RNA virus infection. The authors provide an overview of some of the basic mathematical concepts of mutation rate and quasispecies development. Then, the authors review features of individual RNA virus families and discuss the role of viral quasispecies development in each virus family. 

In general, the topic is important and provides information to the reader about mechanisms of RNA virus evolution. Overall, there are several features of the review that are not well developed. 

The background discussion of RNA virus quasispecies development is not sufficiently developed. The authors do not introduce and define, within the context of genomic organization, the individual contributors of genomic diversity including RDRP proofreading error, genomic recombination, and genomic reassortment. Instead, these are mentioned throughout the manuscript without adequate background information for the reader. Similarly, the authors do not provide enough background and information on the differences between within-host and ecosystem/population quasispecies development. For example, arboviruses RNA genomes undergo both within-host and population-based selection pressures that influence quasispecies development.

The review of individual viral families is vague, superficial, and does not provide any insight into unique family-specific mechanisms that contribute to viral quasispecies development.

The authors briefly mention positive, negative, and purifying selection but do not provide definitions of these important mechanisms and the role they play in RNA virus quasispecies development. In some cases, the authors use the terms incorrectly (ex: line 430). In this context, there is also no discussion of important concepts in the field such as enrichment and selection at regions of immunologic positive selection, the role of evolutionary bottlenecks, and other selective influences.

There are several spelling and grammatical errors throughout the manuscript. 

Author Response

In the manuscript, the authors provide a review of quasispecies development during RNA virus infection. The authors provide an overview of some of the basic mathematical concepts of mutation rate and quasispecies development. Then, the authors review features of individual RNA virus families and discuss the role of viral quasispecies development in each virus family. 

In general, the topic is important and provides information to the reader about mechanisms of RNA virus evolution. Overall, there are several features of the review that are not well developed. 

  • Comment 1:The background discussion of RNA virus quasispecies development is not sufficiently developed. The authors do not introduce and define, within the context of genomic organization, the individual contributors of genomic diversity including RDRP proofreading error, genomic recombination, and genomic reassortment. Instead, these are mentioned throughout the manuscript without adequate background information for the reader. Similarly, the authors do not provide enough background and information on the differences between within-host and ecosystem/population quasispecies development. For example, arboviruses RNA genomes undergo both within-host and population-based selection pressures that influence quasispecies development.

Response: Thank you for the insightful comment. We have added the suggested mechanisms under the section, 1.2 contributors of quasispecies formation.

  • Comment 2: The review of individual viral families is vague, superficial, and does not provide any insight into unique family-specific mechanisms that contribute to viral quasispecies development.

Response: Thank you for pointing out the flaw in the text. We have edited the entire manuscript to remove the redundant parts and also the text which are vague and doesn’t relate to the quasispecies characteristics.

  • Comment 3:The authors briefly mention positive, negative, and purifying selection but do not provide definitions of these important mechanisms and the role they play in RNA virus quasispecies development. In some cases, the authors use the terms incorrectly (ex: line 430). In this context, there is also no discussion of important concepts in the field such as enrichment and selection at regions of immunologic positive selection, the role of evolutionary bottlenecks, and other selective influences.

Response: We have added the details briefly in the section, 1.2 contributors of quasispecies formation. Also, we haven’t put much pressure to provide the details of the mechanism to avoid deviating from the theme of the review. As, the major concern was developing an understanding about the quasispecies, their background and its occurrence in the past pandemics to learn lessons for implementing the knowledge in future antiviral treatment plans.

  • Comment 4: There are several spelling and grammatical errors throughout the manuscript. 

Response: We have thoroughly corrected the manuscript for the grammatical and spelling errors for the best of our knowledge. Text is error free. Moreover, the whole text edited by a native English speaker Dr Brittney Gurda.

Reviewer 2 Report

Thes paper by  Singh and coworkers addresses one very fascinating aspect of virology, i,e, the high variable nature of RNA viral genomes, that generates viral quasispecies, so impacting on the diffusibility and the pathogenesis of these infections. More importantly, this phenomenon challenges the vaccine and therapeutics strategies.

The subject is very topical, but the paper is not suitable for publication in the present version, as it has  numerous  weak points, and, generally, the language is too convoluted, so  it is not linear and easy to follow. In addition, there is no mention of a relevant example of virus quasispecies formation, i.e. HCV, that  is one of the prototypes where the virus quasispecies dynamic models have been formulated, and should be mentioned.

Specific points

Line 68: “The higher the mutation, more will be the probability of extinction” This statement  is not well formulated.  I would suggest: “The higher the number of  mutations that accumulate, the higher  the probability of extinction”

Line 83: is ABOBEC intended to be APOBEC?

Lines 84-87: please include also APOBEC-based mechanisms

Line 117: “A mutation occupies the sequence space and prevails to achieve….”. I would suggest: “A viral genome carrying a   mutation   will occupy the sequence space and will prevail to achieve ……”

Legend to Fig. 4, line 131: “The mutants on the higher peak (yellow) will always  represent low fitness mean in comparison to mutants on flatter peaks (green).” It is not clear why mutants on the higher peaks represent low fitness mean compared to mutants  on flatter peaks. Could you explain better?

Line 144: “accumulation of mutations is advantageous to a virus”. I would modify  “accumulation of mutations is advantageous to a virus provided that this is below the threshold leading to extinction (see above)”.

Lines 173-183: The explanation of the unexpectedly high stability of Paramixoviridae genome, despite its RNA nature, is not convincing. Probably there is no explanation for the unexpected high stability of these RNA genomes, and this shoudl be acknowledged.

Lines 202-205: “The G glycoprotein in these virus shows  the extreme antigenic and sequence divergence, with only 47% amino acid homology between prototype HRSV A and B viruses [27]. It has been hypothesized that changes inamino acid sequences benefits the virus allowing it to escape immunity by modifying its epitopes[28].” The genetic divergence in this case is not attributable to quasispecies, so it is not relevant to the point made by the paper.  Overall, the Paramixoviridae are not relevant for the issue, since   no evidence of quasispecies  formation during their replication is reported. Actually, genetic divergence of these viruses is not related to the intra-host variability, that is mostly relevant for quasispecies formation. I suggest to strongly reduce the paragraph related to this viral family.

Lines 233-242:  High intrinsic  mutation rate and reassortment of genome segments are two distinct phenomena, cooperating  in the overall increase of the variability of influenza virus genome.  This distinction is not clear form the text, and should be addresses in this paragraph.

Lines 305 onward:  HIV. Evolution of HIV quasispecies along disease progression (e.g.R5 vs X4) is at least as important as evolution of drug resistance, but  is not mentioned in thie paper. This additional quasispecies dynamics mechanism should be described.

Lines 382-385: Quotation of ref. 85 (Capobianchi, M.R.; Rueca, M.; Messina, F.; Giombini, E.; Carletti, F.; Colavita, F.; Castilletti, C.; Lalle, E.; Bordi, L.; Vairo, 708 F.J.C.M.; et al. Molecular characterization of SARS-CoV-2 from the first case of COVID-19 in Italy. 2020, 26, 954-956)  is inappropriate, as the description included in the paper refers to the following reference by the same research group: Rueca M, Bartolini B, Gruber CEM, Piralla A, Baldanti F, Giombini E, Messina F, Marchioni L, Ippolito G, Di Caro A, Capobianchi MR. Compartmentalized Replication of SARS-Cov-2 in Upper vs. Lower Respiratory Tract Assessed by Whole Genome Quasispecies Analysis. Microorganisms. 2020 Aug 26;8(9):1302. doi: 10.3390/microorganisms8091302. PMID: 32858978; PMCID: PMC7563410.

Line 404, 407 and 416: Ebola virus, poliovirus and CHIKV. Please, include references for the evidence of quasispecies existance in patients, or quasispecies formation in vitro by these viruses.

492 onwards: Concluding remarks. This chapter is convoluted and not easy to follow. I suggest to make it more linear.

Author Response

This paper by Singh and coworkers addresses one very fascinating aspect of virology, i,e, the high variable nature of RNA viral genomes, that generates viral quasispecies, so impacting on the diffusibility and the pathogenesis of these infections. More importantly, this phenomenon challenges the vaccine and therapeutics strategies.

  • Comment 1:The subject is very topical, but the paper is not suitable for publication in the present version, as it has numerous weak points, and, generally, the language is too convoluted, so it is not linear and easy to follow. In addition, there is no mention of a relevant example of virus quasispecies formation, i.e., HCV, that is one of the prototypes where the virus quasispecies dynamic models have been formulated, and should be mentioned.

Response: Thank you for your insightful comment. We have thoroughly changed the entire manuscript for making it more clear for better understanding. Also, we have mentioned the evidence of quasispecies in Hepatitis C virus under section 3.5.

Specific points

  • Comment 2, Line 68:“The higher the mutation, more will be the probability of extinction” This statement is not well formulated.  I would suggest: “The higher the number of mutations that accumulate, the higher the probability of extinction”.

Response: We agree to the suggestion and made changes at the line 70.

  • Comment 3, Line 83:is ABOBEC intended to be APOBEC?

Response: Thank you for pointing this out. It was a typographical error. We have made the changes at line 111.

  • Comment 4, Lines 84-87: please include also APOBEC-based mechanisms

Response: Thank you for the suggestion. We have added the APOBEC and ADAR based mechanisms at the line 109-119.

  • Comment 5, Line 117:“A mutation occupies the sequence space and prevails to achieve….”. I would suggest: “A viral genome carrying a   mutation   will occupy the sequence space and will prevail to achieve ……”

Response: We agree to the suggestion and made changes at the line 163.

  • Comment 6, Legend to Fig. 4, line 131: “The mutants on the higher peak (yellow) will always represent low fitness mean in comparison to mutants on flatter peaks (green).” It is not clear why mutants on the higher peaks represent low fitness mean compared to mutants on flatter peaks. Could you explain better?

Response: We have incorporated the answer to the question in line 171-186.

  • Comment 7, Line 144:“accumulation of mutations is advantageous to a virus”. I would modify “accumulation of mutations is advantageous to a virus provided that this is below the threshold leading to extinction (see above)”.

Response: We agree to the suggestion and edited the statement at the line198.

  • Comment 8, Lines 173-183: The explanation of the unexpectedly high stability of Paramixoviridae genome, despite its RNA nature, is not convincing. Probably there is no explanation for the unexpected high stability of these RNA genomes, and this should be acknowledged.

Response: We have given the reasons for unexpectedly high stability at the lines 230-237.

  • Comment 9, Lines 202-205: “The G glycoprotein in these virus shows  the extreme antigenic and sequence divergence, with only 47% amino acid homology between prototype HRSV A and B viruses [27]. It has been hypothesized that changes inamino acid sequences benefits the virus allowing it to escape immunity by modifying its epitopes[28].” The genetic divergence in this case is not attributable to quasispecies, so it is not relevant to the point made by the paper.  Overall, the Paramixoviridae are not relevant for the issue, since   no evidence of quasispecies  formation during their replication is reported. Actually, genetic divergence of these viruses is not related to the intra-host variability, that is mostly relevant for quasispecies formation. I suggest to strongly reduce the paragraph related to this viral family.

Response: We agree to the suggestion and revised the paramyxoviridae section for the vague texts and also reduced the paragraph.

  • Comment 10, Lines 233-242: High intrinsic mutation rate and reassortment of genome segments are two distinct phenomena, cooperating in the overall increase of the variability of influenza virus genome.  This distinction is not clear form the text, and should be addresses in this paragraph.

Response: We agree to the suggestion and added briefly about these mechanism in the section 1.2 contributors of quasispecies formation. Also addressed it in the line 273-275 for clarifying the concept.

  • Comment 11, Lines 305 onward:  Evolution of HIV quasispecies along disease progression (e.g.R5 vs X4) is at least as important as evolution of drug resistance, but is not mentioned in the paper. This additional quasispecies dynamics mechanism should be described.

Response: We agree to the suggestion and incorporated supporting text for HIV evolution reflecting at the lines 345-356.

  • Comment 12, Lines 382-385: Quotation of ref. 85 (Capobianchi, M.R.; Rueca, M.; Messina, F.; Giombini, E.; Carletti, F.; Colavita, F.; Castilletti, C.; Lalle, E.; Bordi, L.; Vairo, 708 F.J.C.M.; et al. Molecular characterization of SARS-CoV-2 from the first case of COVID-19 in Italy. 2020, 26, 954-956)  is inappropriate, as the description included in the paper refers to the following reference by the same research group: Rueca M, Bartolini B, Gruber CEM, Piralla A, Baldanti F, Giombini E, Messina F, Marchioni L, Ippolito G, Di Caro A, Capobianchi MR. Compartmentalized Replication of SARS-Cov-2 in Upper vs. Lower Respiratory Tract Assessed by Whole Genome Quasispecies Analysis. Microorganisms. 2020 Aug 26;8(9):1302. doi: 10.3390/microorganisms8091302. PMID: 32858978; PMCID: PMC7563410.

Response: Thank you for the comment. We have changed the reference and updated it.

  • Comment 13, Line 404, 407 and 416: Ebola virus, poliovirus and CHIKV. Please, include references for the evidence of quasispecies existance in patients, or quasispecies formation in vitro by these viruses.

Response: We have added the references for the evidence of the quasispecies formation for Ebola virus, Poliovirus, Chikungunyaon page numbers 11-12 under the section 3.5.

  • Comment 14, 492 onwards: Concluding remarks. This chapter is convoluted and not easy to follow. I suggest to make it more linear.

Response: Thank you for the comment. We have rephrased the concept for more clarity and understanding. Changes are highlighted in lines 596-618.

Round 2

Reviewer 1 Report

Authors have responded to this reviewers comments

Author Response

Response: Thank you for reviewing our manuscript. We have thoroughly corrected the manuscript for the grammatical and spelling errors to the best of our knowledge.

Reviewer 2 Report

The paper is singnificantly improved after the revision. However there are still a number of language imperfections, and in the revised paper several words appear connected to the following ones. I suggest a spelling check and   a minor language revision 

Author Response

Response: Thank you for reviewing our manuscript. We have proofread the manuscript again for spelling mistakes and grammar. The corrections are highlighted in the revised manuscript.